# FEATURE DRIVEN GRAPH COARSENING FOR SCALING GRAPH REPRESENTATION LEARNING

## ABSTRACT

Graphical modelling for structured data analysis has gained prominence across numerous domains. A significant computational challenge lies in efficiently capturing complex relationships within large-scale graph structures. Graph coarsening, which reduces graph size by merging nodes and edges into supernodes and superedges, enhances scalability and is crucial for graph neural networks (GNNs). However, current methods either construct graphs from large-scale attribute data or assume a pre-existing graph before coarsening, limiting their applicability, especially in domains like healthcare and finance where graph structure is often unavailable. In this paper, we present a novel framework that directly learns a coarsened graph from attribute information, reducing computational complexity and enhancing robustness against adversarial attacks, which commonly target vulnerabilities in graph structures. By integrating label information, our framework also enables semi-supervised learning, leading to improved performance on downstream tasks. Extensive experiments show that our method outperforms state-of-the-art coarsening techniques in both accuracy and computational efficiency.

## 1 INTRODUCTION

Graph-based methods are powerful tools for representing relationships between entities and are widely used across domains such as biology, finance, sociology, and engineering. In some cases, relationships are explicit and directly observable, such as friendships in social networks [1], co-authorships in academic publications [2], or neighbouring nodes in sensor networks [3]. However, in many scenarios, such as gene regulation [4], stock trading behavior [4], or drug interactions [5], relationships are latent and must be inferred. Inferring these relationships requires sophisticated domain-specific approaches to construct high-quality graph representations tailored to specific tasks [6; 7; 8; 9].

The increasing scale and diversity of modern datasets [10; 11; 12] pose significant challenges to traditional graph-based methods. Chief among these is the high computational cost of existing algorithms, which becomes prohibitive for large-scale datasets. Techniques such as graph coarsening, condensation, and summarization have been introduced to address scalability by reducing the size of graph structures [13; 14; 15; 16; 17; 18; 19]. However, these approaches rely on the existence of an explicit graph structure, which is not always available. In such cases, a graph must first be inferred from raw data, a computationally intensive process that exacerbates scalability and memory constraints, particularly for large datasets.

To overcome the challenge of learning a graph structure prior to the coarsening process, this work introduces a novel framework that directly learns a coarsened graph from the data itself. By eliminating the need for full graph construction, the proposed method effectively addresses key limitations of traditional approaches, enabling more efficient handling of large-scale datasets and reducing both computational and memory overhead.

We propose a novel optimization-based framework, Coarsened Graph Learning (CGL), which directly learns a coarsened graph from feature data alone. CGL simultaneously resolves the challenges of scalability and the need for an initial graph structure. While GNNs [20; 21; 22] have demonstrated outstanding graph modeling abilities, they are vulnerable to noisy edge graphs, which can significantly degrade their performance [23; 24]. Adversarial edges, often linking nodes with dissimilar labels or attributes, can contaminate node neighborhoods, propagating noise and corrupting node

representations. This issue is particularly prevalent in real-world graphs, such as social networks, where bots create links with regular users to spread misinformation. Considerable research has focused on developing robust GNNs [25; 26], and our method also offers robustness by learning a coarsened graph independently of the graph structure, making it impervious to adversarial attacks on the structure.

The proposed CGL framework is a multi-block, non-convex optimization problem that can be efficiently solved using the Block Successive Upper-bound Minimization (BSUM) technique, where variables are updated iteratively while keeping others fixed. To evaluate our model, we compare the node classification performance of CGL and its semi-supervised variant, SCGL, against state-of-the-art methods such as GCOND [15], SCAL [16], and FGC [13] on both homophilic and heterophilic datasets. We also assess the time required for graph learning and coarsening, comparing traditional methods like [6] with our direct coarsened graph learning approach via CGL. Finally, we demonstrate the robustness of CGL by highlighting its resilience to adversarial attacks on graph structure, where traditional coarsening methods falter.

Additionally, in practical applications, data points often come with some node label information. We integrate this label information into our objective function, significantly enhancing downstream task performance. Extensive experiments demonstrate that incorporating label information into the CGL objective substantially improves its overall effectiveness. The key contributions of this work are:

1. A pipeline for performing downstream tasks using GNNs by utilizing only the features.

2. A novel optimization-based framework that *"learns a coarsened graph directly from features"* or raw data for scaling of GNNs.

3. A label informed *"semi-supervised coarsened graph learning"* framework which enhances the downstream task performance using GNNs.

4. Through experimentation demonstrating the efficacy of proposed frameworks in terms of node classification performance and computational efficiency.

## 2 BACKGROUND AND PROBLEM FORMULATION

In this section, we review the basics of graphs, learning from graph data, graph coarsening, and propose a formulation for directly learning a coarsened graph from raw data only. We represent graph as $\mathcal{G}(V, E, X, Y)$, where $V$ is the vertex set comprising the individual nodes constituting the graph, and $E$ is the edge set capturing the relationships or connections between these nodes. Next, $X \in \mathbb{R}^{p \times n}$ is the feature matrix, where $X = [\mathbf{x}_1, \mathbf{x}_2, \ldots, \mathbf{x}_p]^\top$. Each $\mathbf{x}_i \in \mathbb{R}^n$ represents the feature associated with node $i$. Finally, $Y$ represents the label information available for some of the nodes of the graph. Graphs are generally represented by the Laplacian matrix or the Adjacency matrix. A matrix is said to be a combinatorial Laplacian matrix if it belongs to the following set [27; 28]:

$$\mathcal{S}_\Theta = \Big\{ \Theta_{ij} = \Theta_{ji} \leq 0 \text{ for } i \neq j; \Theta_{ii} = -\sum_{j \neq i} \Theta_{ij} \Big\}. \tag{1}$$

Both $\Theta$ and $A$ represent the same graph and are linear transformations of each other $A_{ij} = -\Theta_{ij}$ for all $i \neq j$, and $A_{ii} = 0$ for all $i$. However, the Laplacian matrix is positive semidefinite has zero row sum and column sum. Laplacian matrix representation has been well recognized as a tool for embedding, manifold learning, spectral sparsification, clustering and semi-supervised learning.

### 2.1 GRAPH LEARNING FROM DATA $X$

How to learn a $p$ nodes graph with $p$ data points $X = [\mathbf{x}_1, \mathbf{x}_2, \ldots, \mathbf{x}_p]^\top \in \mathbb{R}^{p \times n}$ is well understood [28; 29; 30; 6; 31]. The goal here is to infer the connectivity relationships between pair of datapoints $\mathbf{x}_i$ and $\mathbf{x}_j$ in form of a graph matrix which is generally done by solving the following class of optimization problem:

$$\min_{\Theta \in \mathcal{S}_\Theta} f(\Theta) + g(X, \Theta) + \alpha h(\Theta) \tag{2}$$

In this context, $f(\Theta)$ is a function that ensures the connectivity of the learned graph. For example, $f(\Theta) = -\log \det(\Theta + J)$ has been used in [32], and $f(\Theta) = -\mathbf{1}^T \log(A\mathbf{1})$, where $A = D - \Theta$

is the adjacency matrix and $D$ is the degree matrix, has been employed in [6]. These functions are designed to ensure connectivity in the graph. Next, the term $g(X, \Theta)$ is a graph fitting term. One common choice is $g(X, \Theta) = \text{tr}(X^\top \Theta X)$ [33; 6; 32], known as the smoothness or Dirichlet energy of the graph. Minimizing this term implies that the learned graph is smooth, meaning that nodes with similar features are connected through stronger weights. Lastly, $h(\Theta)$ acts as a regularizer to impose additional desirable properties on the graph. For instance, $h(\Theta) = \|\Theta\|_F^2$ is often used to ensure sparsity in the learned graph.

## 2.2 Graph Neural Network

Graph Neural Networks [20; 21] (GNNs) are highly effective at capturing the representations of nodes and edges within a graph. GNNs are designed to perform various tasks e.g. node classification, graph classification, edge prediction, etc. on graph data $\mathcal{G}(V, E, X, Y)$. GNNs utilize message-passing mechanisms where each node aggregates information from its neighbors to update its representation iteratively. One of the common architecture of GNN is graph convolution network(GCN). Let $h_v^t$ represent the feature vector of node $v$ at iteration $t$. The representation $h_v^{t+1}$ is computed by aggregating the representation vectors $h_u^t$ of neighboring nodes $u$, resulting in a more stable and informative representation. These enhanced representation vectors are then input into a task-specific multi-layer perceptron (MLP) for final predictions.

$$h_v^{t+1} = Aggregate(\{h_v^t\} \cup \{h_u^t : u \in Neighbour(v)\}) \tag{3}$$

where $h_v^t$ is the representation vector of node $v$ after $t$ iterations. To perform downstream tasks using GNNs, the Laplacian matrix, feature matrix, and some node label information are needed. For various datasets, this information is available; however, for some datasets, the graph matrix is not available, and can be learned the graph using graph learning techniques, see for a recent work in [34].

## 2.3 Graph Dimensionality Reduction

The size of the dataset is increasing day by day, leading to significant computational costs and memory requirements to learn the graph. Graph condensation and coarsening [13; 14; 15; 16; 17; 18; 19] are techniques to reduce graph complexity while preserving key information. In graph condensation [15], a smaller synthetic graph $G_c = (\mathcal{V}_c, \mathcal{E}_c, \mathbf{X}_c, \mathbf{Y}_c)$ is generated to ensure that a graph neural network (GNN) trained on $G_c$ performs comparably to one trained on the original $G$, using a loss function $L(GNN_{\theta_{G_c}}(G_c), GNN_{\theta_G}(G))$. Graph coarsening [13; 14; 19], on the other hand, maps $G = (\mathcal{V}, \mathcal{E})$ to a coarser graph $G_c = (\mathcal{V}_c, \mathcal{E}_c)$, with $|\mathcal{V}_c| < |\mathcal{V}|$, often through surjective mappings $P$, where each non zero entry of mapping matrix $P$ i.e. $P_{ij}$ indicates the $j-$th node of the original graph got mapped to the $i$-th supernode of the coarsened graph. While condensation focuses on preserving GNN performance, coarsening emphasizes maintaining structural properties. However, these techniques assume the graph is available, which is not the case in domains like gene regulation or financial stock analysis. In such cases, graph learning is required as a preliminary step, which is computationally intensive for large datasets, and performing coarsening afterward further increases complexity. To address this, we have developed a coarsened graph learning technique that directly learns a coarsened graph from raw data, eliminating the need for separate graph learning and coarsening steps, thereby significantly reducing computational overhead.

**Goal:** *Given $p$ data points $X = [\mathbf{x}_1, \mathbf{x}_2, \ldots, \mathbf{x}_p]^\top$ the goal is to efficiently learn a good quality coarsened graph $\mathcal{G}_c(\Theta_c, X_c)$ with $k$ nodes, where $k \ll p$.*

## 2.4 Proposed Formulation for Coarsened Graph Learning(CGL)

Consider an toy example shown in the Figure 1 for learning a coarsened graph from raw data $X$ that consists of 10 data points. Data points $\{x_1, x_3, x_8\}$ are mapped to $\tilde{x}_1$; $\{x_4, x_6\}$ are mapped to $\tilde{x}_4$; $\{x_2, x_9, x_{10}\}$ are mapped to $\tilde{x}_3$; and $\{x_5, x_7\}$ are mapped to $\tilde{x}_2$. Utilizing coarsened graph features $\tilde{X}$, we learn a coarsened graph. In our framework, we simultaneously learn the coarsened graph features $\tilde{X}$ and the Laplacian matrix $\Theta_c$. The original graph feature matrix $X$ is mapped to coarsened graph feature matrix $\tilde{X}$ using the relation $\tilde{X} = PX$, where $P$ is the mapping matrix, for this toy example the mapping matrix $P$ is shown in below figure.

Next, each entry of $P \in \mathbb{R}_+^{p \times k}$ matrix i.e, $P_{ij}$ indicate that the $j$-th original data point is mapped to $i$-th supernode of coarsened graph. Where $p$ is the number of original data points and k is the

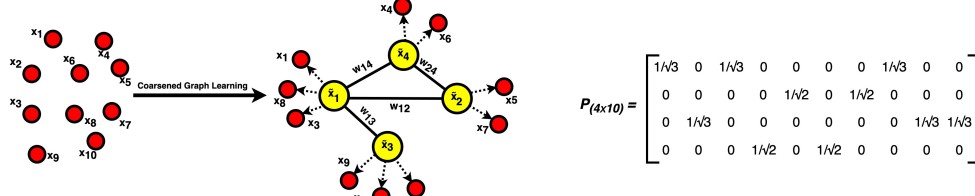

Figure 1: Given the data $X$, we aim to learn a coarsened graph $\mathcal{G}_c(\Theta_C, \tilde{X})$. Each non-zero entry of mapping matrix $P \in \mathbb{R}_+^{k \times p}$ i.e. $P_{ij}$ signifies that $j^{th}$ data point get mapped to $i^{th}$ supernode of the coarsened graph.

number of nodes or supernodes of the coarsened graph. For a balanced mapping the mapping matrix $P$ should belong to the following set:

$$\mathcal{P} = \left\{ P \geq 0 \mid \langle [P^\top]_i, [P^\top]_j \rangle = 0 \; \forall \; i \neq j, \langle P_i, P_i \rangle = I, \|[P^\top]_i\|_0 \geq 1 \text{ and } \|P_i\|_0 = 1 \right\} \quad (4)$$

Given the data $X \in \mathbb{R}^{p \times n}$, the proposed formulation for learning a coarsened graph $\mathcal{G}_c(\Theta_c \in \mathbb{R}^{k \times k}, \tilde{X} \in \mathbb{R}^{k \times n})$ where $k << p$ is:

$$\begin{aligned} \underset{\Theta_c, \tilde{X}, P}{\text{minimize}} \quad & -\gamma \log \det(\Theta_c + J) + \text{tr}(\tilde{X}^T \Theta_c \tilde{X}) + h(\Theta_c) \\ \text{subject to} \quad & \tilde{X} = PX, \; P \in \mathcal{S}_p, PP^T = I \end{aligned} \quad (5)$$

where $\mathcal{S}_p = \left\{ P \geq 0 \mid \|[P^T]_i\|_2^2 \leq 1 \; \forall \; i = 1, .., k \right\}$. where $P$ denotes the mapping matrix. The term $-\gamma \log \det(\Theta_c + J)$ ensures that the learned coarsened graph is connected. By minimizing $\text{tr}(\tilde{X}^T \Theta_c \tilde{X})$, we ensure that the coarsened graph is smooth. Here, $\tilde{X} = PX$ transforms the original data $X$ into the feature matrix $\tilde{X}$ of the coarsened graph. The function $h(\Theta_c)$ acts as a regularizer to impose a specific structure on the coarsened graph; for example, $h(\Theta_c) = \|\Theta_c\|_F^2$ encourages sparsity within the learned coarsened graph.

## 3 CGL ALGORITHM DEVELOPMENT

Before moving towards the algorithm development, we will discuss a linear and adjoint operator $\mathcal{L}$ and $\mathcal{L}^*$.

**Definition 1** *The linear operator $\mathcal{L} : \mathbb{R}^{p(p-1)/2} \to \mathbb{R}^{p \times p}$, $\mathbf{w} \mapsto \mathcal{L}\mathbf{w}$, is defined as [28]*

$$[\mathcal{L}\mathbf{w}]_{ij} = \begin{cases} -\mathbf{w}_{i+d_j} & \text{for } i > j; \quad [\mathcal{L}\mathbf{w}]_{ji} \text{ for } i < j; \quad -\sum_{i \neq j}[\mathcal{L}\mathbf{w}]_{ij} \text{ for } i = j, \end{cases} \quad (6)$$

*where $d_j = -j + \frac{j-1}{2}(2p - j)$. The adjoint operator $\mathcal{L}^* : \mathbb{R}^{p \times p} \to \mathbb{R}^{p(p-1)/2}$, $Y \mapsto \mathcal{L}^* Y$, is defined by [28]:*

$$[\mathcal{L}^* Y]_k = Y_{i,i} - Y_{i,j} - Y_{j,i} + Y_{j,j}, \quad k = i - j + \frac{j-1}{2}(2p - j), \quad (7)$$

*where $i, j \in \mathbb{Z}^+$ satisfy $k = i - j + \frac{j-1}{2}(2p - j)$ and $i > j$.*

### 3.1 COARSENED GRAPH LEARNING(CGL)

Utilizing the Laplacian operator defined in 6, given the data $X \in \mathbb{R}^{p \times n}$ the proposed formulation equation 5, can be reformulated as:

$$\begin{aligned} \underset{\mathbf{w}, \tilde{X}, P}{\text{minimize}} \quad & -\gamma \log \det(\mathcal{L}\mathbf{w} + J) + \text{tr}(\tilde{X}^T \mathcal{L}\mathbf{w}\tilde{X}) + \frac{\beta}{2}\|\mathcal{L}\mathbf{w}\|_F^2 \\ \text{subject to} \quad & P \in \mathcal{S}_p, \quad PP^T = I, \quad \tilde{X} = PX, \quad \mathbf{w} \geq \mathbf{0} \end{aligned} \quad (8)$$

where $\mathcal{S}_p = \left\{ P \geq 0 \middle| \, \|[P^\top]_i\|_2^2 \leq 1 \, \forall \, i = 1, .., k \right\}$ is a closed convex set. We further relax the problem 8 by incorporating the term $\frac{\delta}{4}\|PP^T - I\|_F^2$ and $\frac{\alpha}{2}\|\tilde{X} - PX\|_F^2$ with $\delta > 0$ and $\alpha > 0$ instead of addressing the constraints $PP^T = I$ and $\tilde{X} = PX$. Now the problem 8 can be reformulated as:

$$\begin{aligned} \underset{\mathbf{w} \geq \mathbf{0}, \tilde{X}, P \in \mathcal{S}_p}{\text{minimize}} \quad & f_{CGL} = -\gamma \log \det(\mathcal{L}\mathbf{w} + J) + \text{tr}(\tilde{X}^T \mathcal{L}\mathbf{w}\tilde{X}) + \frac{\delta}{4}\|PP^T - I\|_F^2 \\ & + \frac{\alpha}{2}\|\tilde{X} - PX\|_F^2 + \frac{\beta}{2}\|\mathcal{L}\mathbf{w}\|_F^2 \end{aligned} \tag{9}$$

The problem 9 is a multi block non convex optimization problem. We solved this problem using the BSUM framework [35], which involves updating one variable at a time while keeping the others fixed. Collecting the variables as $\mathbf{w} \in \mathbb{R}_+^p, \tilde{X} \in \mathbb{R}^{k \times n}, P \in \mathbb{R}_+^{k \times p}$, we create a block MM-based algorithm that updates one variable at a time while keeping the others constant.

**Update of w:** Treating $P$ and $\tilde{X}$ fixed, with $\mathbf{w}$ considered as the variable, the corresponding subproblem for optimizing $\mathbf{w}$ can be expressed as follows:

$$\underset{\mathbf{w} \geq \mathbf{0}}{\text{minimize}} f(\mathbf{w}) = -\gamma \log \det(\mathcal{L}\mathbf{w} + J) + \text{tr}(\tilde{X}^T \mathcal{L}\mathbf{w}\tilde{X}) + \frac{\beta}{2}\|\mathcal{L}\mathbf{w}\|_F^2 \tag{10}$$

The function $f(\mathbf{w})$ is a strictly convex function in $\mathbf{w}$ [28]. To get the closed form solution of problem equation 10, using first order taylor series expansion, we majorized $f(\mathbf{w})$ at $\mathbf{w}^t$ by the function [36; 37; 35]:

$$g\left(\mathbf{w}|\mathbf{w}^{(t)}\right) = f\left(\mathbf{w}^{(t)}\right) + \left(\mathbf{w} - \mathbf{w}^{(t)}\right)^\top \nabla f\left(\mathbf{w}^{(t)}\right) + \frac{L}{2}\|\mathbf{w} - \mathbf{w}^{(t)}\|^2, \tag{11}$$

where $f(\mathbf{w})$ is $L-$Lipschitz continuous gradient function having $L = \max(L_1, L_2, L_3)$ with $L_1, L_2, L_3$ the Lipschitz constants of $-\gamma \log \det(\mathcal{L}\mathbf{w} + J)$, $\text{tr}(\tilde{X}^T \mathcal{L}\mathbf{w}\tilde{X})$, and $\frac{\beta}{2}|\mathcal{L}\mathbf{w}|_F^2$ respectively. After ignoring the constant term, the majorised problem of equation 10 is

$$\underset{\mathbf{w} \geq 0}{\text{minimize}} \quad \frac{1}{2}\mathbf{w}^\top \mathbf{w} - \mathbf{w}^T \mathbf{a} \tag{12}$$

where, $\mathbf{a} = \mathbf{w}^t - \frac{1}{L}\nabla f(\mathbf{w}^t)$ and, $\nabla f(\mathbf{w}^t) = -\frac{\gamma}{\beta}\mathcal{L}^*(\mathcal{L}\mathbf{w}^t + J)^{-1} + \mathcal{L}^*(\mathcal{L}\mathbf{w}^t) + \mathcal{L}^*(\frac{1}{\beta}\tilde{X}\tilde{X}^T)$. Using KKT optimality condition, the solution of problem equation 12 is

$$\mathbf{w}^{(t+1)} = \left(\mathbf{w}^{(t)} - \frac{1}{L_1}\nabla f\left(\mathbf{w}^{(t)}\right)\right)^+, \tag{13}$$

where $x^+ = max(x, 0)$

**Update of $P$:** While updating $P$, treating $\mathbf{w}$ and $\tilde{X}$ as fixed and $P$ as variable, the subproblem for $P$ is:

$$\underset{P \geq 0}{\text{minimize}} \quad \frac{\alpha}{2}\|\tilde{X} - PX\|_F^2 + \frac{\delta}{4}\left\|PP^T - I\right\|_F^2 \tag{14}$$

The Lagrangian function of equation 14 by introducing the lagrangian multiplier $\phi$ is:

$$L(P, \phi) = \frac{\alpha}{2}\left\|\tilde{X} - PX\right\|_F^2 + \frac{\delta}{4}\left\|PP^T - I\right\|_F^2 - tr(\phi^T P) \tag{15}$$

The derivative Lagrangian function w.r.t. $P$ and equate it to zero, we get:

$$\frac{\partial L(P, \phi)}{\partial P} = -\alpha(\tilde{X} - PX)X^T + \delta(PP^T - I)P - \phi = 0 \tag{16}$$

To get the update rule of $P$ matrix, the karush-kuhn-Tucker(KKT) condition $\phi_{ij}P_{ij} = 0$ is applied. We get:

$$P_{ij} \leftarrow P_{ij} \frac{(\alpha PXX^T + \delta PP^T P)_{ij}}{(\alpha \tilde{X}X^T + \delta P)_{ij}} \tag{17}$$

**Update of $\tilde{X}$:** Treating $\mathbf{w}$ and $P$ fixed and $\tilde{X}$ as variable. The subproblem for updating $\tilde{X}$ is

$$\underset{\tilde{X}}{\text{minimize}} \quad \text{tr}(\tilde{X}^T \mathcal{L}\mathbf{w}\tilde{X}) + \frac{\alpha}{2}\left\|\tilde{X} - PX\right\|_F^2 \tag{18}$$

The closed form solution of problem equation 18 can be obtained by putting the gradient of $f(\tilde{X})$ to zero.

$$\tilde{X} = \alpha(2\mathcal{L}\mathbf{w} + \alpha I)^{-1}PX \tag{19}$$

---

**Algorithm 1: CGL Algorithm**

**Input:** $X, \alpha, \beta, \gamma, \delta, P^0, \mathbf{w}^0, \tilde{X}^0 = P^0 X$
$t \leftarrow 0$;
**while** *stopping criteria not met* **do**
  Update $\mathbf{w}^{t+1}$, $P^{t+1}$ and $\tilde{X}^{t+1}$ as in equation 13, equation 17 and equation 19 respectively.
  $t \leftarrow t + 1$;
**end**
**Output:** $P$, $\mathbf{w}$, and $\tilde{X}$

---

The worst-case computational complexity of CGL alorithm per iteration is $\mathcal{O}(kpn + k^2 p)$.

**Theorem 1** *The sequence $\{\mathbf{w}^{(t)}, P^{(t)}, \tilde{X}^{(t)}\}$ generated by Algorithm 1 converges to the set of Karush–Kuhn–Tucker (KKT) points of Problem equation 9.*

*Proof:* The proof is deferred to the Appendix A.1.

# 4 SEMI-SUPERVISED COARSENED GRAPH LEARNING (SCGL)

In general, datasets consist of features $X$ and label information $Y$, where $Y \in \{0,1\}^{n \times l}$. If node $v^i$ is labeled, $\mathbf{y}_i$ represents the corresponding one-hot indicator vector; otherwise, $\mathbf{y}_i = 0$ for unlabeled data. Incorporating this label information while learning a coarsened graph may lead to improve performance in downstream tasks. Consider a toy example with 10 data points and 3 labels: {red, blue, green}. We aim to learn a coarsened graph with 4 nodes. Various structures for the coarsened graph are possible. Here, we illustrate two possible coarsened graphs, $\mathcal{G}_{c1}$ and $\mathcal{G}_{c2}$. Training a GNN on $\mathcal{G}_{c2}$ results in a better-trained model compared to $\mathcal{G}_{c1}$. This is because, in $\mathcal{G}_{c2}$, data points with similar labels are grouped into the same supernode, leading to a sparser $[PY]$ matrix, which enhances the training efficiency and effectiveness of the GNN.

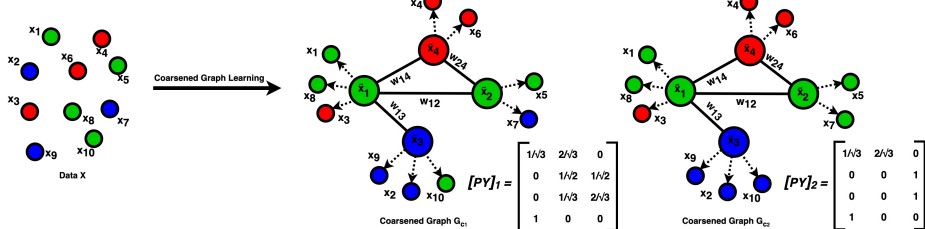

Figure 2: This illustration demonstrates that, for a given original graph $\mathcal{G}$, consider two coarsened graphs, $\mathcal{G}_{c1}$ and $\mathcal{G}_{c2}$. In $\mathcal{G}_{c2}$, data points with similar labels are mapped to the same supernode, resulting in a sparse $PY$ matrix. This sparsity leads to more effective training of the GNN compared to $\mathcal{G}_{c1}$.

Considering the function $\mathcal{G}(P, Y)$ that integrates the label $Y$ in the optimization framework for learning a coarsened graph, which introduces sparsity in the $[PY]$ matrix and ensures that nodes with similar labels are grouped into the same supernode, the proposed formulation for learning a coarsened graph given $(X, Y)$ is as follows:

$$\begin{aligned}
\underset{\mathbf{w} \geq \mathbf{0}, \tilde{X}, P}{\text{minimize}} \quad & f_{CGL} + \eta g(P, Y) \\
\text{subject to} \quad & P \in \mathcal{S}_p
\end{aligned} \tag{20}$$

There are various possibilites of $g(P, Y)$ that introduces sparsity in the $[PY]$ matrix, here we consider $\|PY\|_F^2$. Each entry of the matrix $[PY] \in \mathbb{R}_+^{k \times l}$, denoted as $[PY]_{ij}$ quantifies the number of data points with the $j$-th label is mapped to the $i$-th supernode of the coarsened graph. For an optimal coarsened graph, each row $i$ of this matrix should contain exactly one non-zero entry, which should be 1. This indicates that all data points mapped to the $i$-th supernode have only the $j$-th label. The algorithm development of SCGL follows that of the CGL algorithm. The key difference is that in the update rule for $P$, $\frac{\eta}{2}PYY^\top$ is added to the numerator.

## 5 EXPERIMENTS

In this section, we conduct an empirical evaluation to assess the effectiveness of our proposed algorithms. We validate the CGL and SCGL algorithms through a comprehensive set of experiments on real datasets both homophilic and heterophilic graphs, comparing them to state-of-the-art approaches such as FGC [14], GCOND [15], and SCAL [16].

### 5.1 EXPERIMENTAL SETUP

**Dataset**: We run experiments on ten real world datasets including homophilic and heterophilic datasets. More details about the dataset are in the Appendix A.2.

**Experimental Settings:** Given a graph $\mathcal{G}(\Theta, X, Y)$, we performed node classification task experiments in three settings:

**Set1**: Use $X \in \mathbb{R}^{p \times k} \xrightarrow[via\ 2]{Learn} \mathcal{G}(\Theta_l) \xrightarrow[on\ \mathcal{G}(\Theta_l, X, Y)]{Train\ GNNs} \Rightarrow$ Testing on $\mathcal{G}(\Theta, X, Y)$.

**Set2**: Using $X$ or $(X, Y) \xrightarrow[proposed\ methods]{Learn\ via} \mathcal{G}_c(\Theta_c, \tilde{X}, \tilde{Y}) \xrightarrow[on\ \mathcal{G}_c(\Theta_c, \tilde{X}, \tilde{Y})]{Train\ GNNs}$ Testing on $\mathcal{G}(\Theta, X, Y)$.

**Set3**: For the existing baseline methods, we performed the node classification task in two ways:

(a): Given $\mathcal{G}(\Theta, X, Y) \xrightarrow[via\ baselines]{Learn} \mathcal{G}_c(\Theta_c, \tilde{X}, \tilde{Y}) \xrightarrow[on\ \mathcal{G}_c(\Theta_c, \tilde{X}, \tilde{Y})]{Train\ GNNs}$ Testing on $\mathcal{G}(\Theta, X, Y)$

(b): Using $X \xrightarrow[via\ 2]{Learn} \mathcal{G}(\Theta_l) \xrightarrow[via\ baselines]{Learn} \mathcal{G}_c(\Theta_c, \tilde{X}, \tilde{Y}) \xrightarrow[on\ \mathcal{G}(\Theta_c, \tilde{X}, \tilde{Y})]{Train\ GNNs}$ Testing on $\mathcal{G}_c(\Theta, X, Y)$

However, note that our proposed algorithms, CGL and SCGL, do not utilize the given adjacency matrix $\Theta$ of the original graph. We have only used the feature matrix $X$ or $(X, Y)$ to learn the coarsened graph. The adjacency matrix $\Theta$ is utilized only during testing on the original graph.

| Dataset | r | Baseline | | | Proposed | | Whole Dataset |
|---------|---|----------|---|---|----------|---|---------------|
| | | GCOND | SCAL | FGC | CGL | SCGL | |
| Cora | 0.3 | 81.56±0.6 | 79.42±1.71 | 85.79±0.24 | 87.77±0.92 | **89.29±0.68** | |
| | 0.1 | 81.37±0.40 | 71.38±3.62 | 81.46±0.79 | 82.60±0.15 | **88.66±1.14** | 89.5 ±1.23 |
| | 0.05 | 79.93±0.44 | 55.32±7.03 | 80.01±0.51 | 79.02±0.06 | **88.07±0.59** | |
| Citeseer | 0.3 | 72.43±0.94 | 68.87±1.37 | 74.64±1.37 | 77.15±0.09 | **79.59±0.44** | |
| | 0.1 | 70.46±0.47 | 71.38±3.62 | 73.36±0.53 | 74.03±0.24 | **79.17±0.54** | 78.09±1.96 |
| | 0.05 | 64.03±2.4 | 55.32±7.03 | 71.02±0.96 | 70.60±0.23 | **79.68±0.50** | |
| Pubmed | 0.05 | 78.16±0.30 | 72.82±2.62 | 80.73±0.44 | 84.73±0.21 | **86.43±0.09** | |
| | 0.03 | 78.04±0.47 | 70.24±2.63 | 79.91±0.30 | 81.40±0.66 | **85.70±0.11** | 88.89±0.59 |
| | 0.01 | 77.2±0.20 | 54.49±10.4 | 78.42±0.43 | 80.77±0.02 | **84.73±0.41** | |
| Co-phy | 0.05 | 93.05±0.26 | 73.09±7.41 | 94.27±0.25 | 94.76±0.05 | **95.32±0.21** | |
| | 0.03 | 92.81±0.31 | 63.65±9.65 | 94.02±0.20 | 91.66±0.48 | **94.61±0.12** | 96.22±0.72 |
| | 0.01 | 92.79±0.4 | 31.08±2.65 | 93.08±0.22 | 90.65±0.04 | **94.48±0.10** | |
| Co-cs | 0.05 | 86.29±0.63 | 34.45±10.1 | 88.06±0.78 | 88.69±0.16 | **91.47±0.13** | |
| | 0.03 | 86.32±0.45 | 26.06±9.29 | 87.23±0.70 | 88.32±4.38 | **90.61±0.29** | 93.32±0.60 |
| | 0.01 | 84.01±0.02 | 14.42±8.5 | 84.71±2.12 | 86.60±0.32 | **87.30±0.76** | |
| Flickr | 0.005 | 47.14±0.32 | OOM | 45.32±0.49 | 55.87±0.23 | **57.23±0.17** | |
| | 0.003 | 46.81±0.54 | OOM | 46.23±0.33 | 49.34±0.68 | **50.40±0.64** | 61.26±0.30 |
| | 0.001 | 46.54±0.72 | OOM | 42.55±1.57 | 47.32±0.19 | **49.56±0.44** | |

Table 1: This table shows the node classification performance of the proposed algorithms CGL and SCGL compared to state-of-the-art methods GCOND [15], SCAL [16], and FGC [13] on homophilic datasets. It is evident that for coarsening, we have utilized only the feature matrix $X$ and $(X, Y)$ for CGL and SCGL respectively, and yet our framework outperforms the existing state-of-the-art graph coarsening methods, which consider both the Laplacian and the feature matrix for coarsening. It is important to note that for SCGL, we have only considered labels that existing graph coarsening techniques utilized during the training of GNNs.

| Dataset | r | Baseline | Proposed | | |
| | | FGC | CGL | SCGL | Whole Dataset |
|---|---|---|---|---|---|
| Genius | 0.001 | 74.21±2.35 | 78.30±0.67 | **80.01±0.78** | 87.42±0.53 |
| Arxiv-year | 0.001 | 27.2±0.54 | 27.42±0.88 | **31.86±0.38** | 46.02±0.33 |
| OGBN-Proteins | 0.001 | 65.2±0.74 | 66.93±0.70 | **70.40±0.82** | 72.51±0.12 |
| OGBN-Products | 0.001 | 62.4±0.43 | 67.52±0.12 | **70.12±0.23** | 82.33±0.22 |

Table 2: Performance evaluation of the proposed framework on large-scale datasets with a coarsening ratio of 0.001. Comparisons were made against FGC, as other baselines ran out of memory during evaluation. The results demonstrate that the proposed framework significantly enhances the downstream GCN performance compared to the state of the art techniques.

| Dataset | r | Baseline | Proposed | |
| | | FGC | SCGL | Whole Dataset |
|---|---|---|---|---|
| Xin | 0.5 | 90.34±0.32 | **93.92±0.67** | |
| | 0.3 | 90.12±0.84 | **92.89±1.12** | 95.58 ±0.23 |
| | 0.1 | 90.01±0.44 | **90.68±0.13** | |

Table 3: Node classification accuracy on genome sequence data where graph structure matrix is not explicitly given.

| Dataset | r | GCOND ($\tau$) | FGC ($\tau$) | SCGL ($\tau$) | Wh. Data |
|---|---|---|---|---|---|
| Cora | 0.05 | 329.86 | 4.84 | **0.41** | 2.86 |
| Citeseer | 0.05 | 333.46 | 7.07 | **0.64** | 5.24 |
| Pubmed | 0.01 | 1934.56 | 34.48 | **1.58** | 58.85 |
| Genius | 0.001 | OOM | 2452 | **20.21** | 892 |
| Arxiv-year | 0.001 | OOM | 1765 | **27.16** | 635 |
| OGBN-Proteins | 0.001 | OOM | 10367 | **36.39** | 9654 |

Table 4: This table shows time($\tau$) in seconds to perform coarsening and classification and shows that the proposed SCGL technique is significantly faster than existing graph coarsening techniques. Methods like FGC [13] and GCOND [15], which consider the adjacency matrix during coarsening, require substantial time, defeating the purpose of coarsening. In contrast, focusing solely on features to learn a coarsened graph significantly reduces the time required. For the **OGBN-protein** dataset, it is almost 30 times faster, and for the **Genius** dataset, it is almost 40 times faster than the existing state-of-the-art method.

## 5.2 NODE CLASSIFICATION

To validate the efficacy of our proposed algorithm, we performed node classification tasks on both homophilic and heterophilic datasets. As this is the first study to learn a coarsened graph directly from the raw dataset $X$, there are no existing benchmark datasets. We used the graph dataset $\mathcal{G}(X, \Theta, Y)$ for this purpose. To derive the coarsened graph $\mathcal{G}_c(\Theta_c, \tilde{X})$, we utilized the feature matrix $X$ and $(X, Y)$ through the CGL and SCGL algorithms, respectively, where $Y$ is the label information for some of the data points. After obtaining the coarsened graph, we determined the labels of the coarsened graph using the relation $\tilde{Y} = \arg\max(PY)$. We then trained a graph neural network using the coarsened graph $\mathcal{G}_c(\Theta_c, \tilde{X}, \tilde{Y})$ and tested it on the original graph. It is important to note that the original graph Laplacian matrix was used only during testing. The coarsened graph was learned solely from the feature matrix $X$. Also. we have taken standard split ratio of $(80\%, 10\%, 10\%)$ where $80\%$ for training, $10\%$ for validation, and $10\%$ for testing. We begin by evaluating our coarsening framework on traditional small- and medium-scale graph datasets, with results presented in Table 1, where all baseline methods are included for comparison. To further assess scalability, we conduct

evaluations on very large-scale datasets. In this setting, all baseline coarsening methods, except FGC, encountered memory limitations, and the results are reported in Table 2. To demonstrate the real-world applicability of our framework, which directly learns a coarsened graph from the feature matrix, we utilize the Xin dataset. This dataset comprises transcriptomes from single-cell RNA sequencing (scRNA-seq) of pancreatic cells, covering 1,449 cells and the expression profiles of 33,889 genes across four primary pancreatic cell types: alpha, beta, delta, and gamma. The evaluation results are detailed in Table 3. Additional results on heterophily datasets is provided in Appendix A.3.

| Dataset | r | FGC | FGCL | CGL | SCGL | Wh.Data | L.Data |
|---------|------|------------|------------|------------|-----------------|------------|--------|
| Cora | 0.3 | 85.79±0.24 | 74.56±0.32 | 87.77±0.92 | **89.29±0.68** | | |
| | 0.1 | 81.46±0.79 | 68.68±0.58 | 82.60±0.15 | **88.66±1.14** | 89.5±1.23 | 92.54 |
| | 0.05 | 80.01±0.51 | 62.85±0.44 | 79.02±0.06 | **88.07±0.59** | | |
| Citeseer | 0.3 | 74.64±1.37 | 75.47±0.85 | 77.15±0.09 | **79.59±0.44** | | |
| | 0.1 | 73.36±0.53 | 69.88±0.09 | 74.03±0.24 | **79.17±0.54** | 78.09±1.96 | 84.58 |
| | 0.05 | 71.02±0.96 | 67.26±0.52 | 70.60±0.23 | **79.68±0.50** | | |
| Pubmed | 0.05 | 80.73±0.44 | 79.00±0.68 | 84.73±0.21 | **86.43±0.09** | | |
| | 0.03 | 79.91±0.30 | 78.92±0.78 | 81.40±0.66 | **85.70±0.11** | 88.89±0.59 | 88.81 |
| | 0.01 | 78.42±0.43 | 77.23±0.52 | 80.77±0.02 | **84.73±0.41** | | |
| Co-phy | 0.05 | 94.27±0.25 | | 94.76±0.05 | **95.32±0.21** | | |
| | 0.03 | 94.02±0.20 | OOM | 91.66±0.48 | **94.61±0.12** | 96.22±0.72 | OOM |
| | 0.01 | 93.08±0.22 | | 90.65±0.04 | **94.48±0.10** | | |
| Co-cs | 0.05 | 88.06±0.78 | | 88.69±0.16 | **91.47±0.13** | | |
| | 0.03 | 87.23±0.70 | OOM | 88.32±4.38 | **90.61±0.29** | 93.32±0.60 | OOM |
| | 0.01 | 84.71±2.12 | | 86.60±0.32 | **87.30±0.76** | | |

Table 5: This table presents node classification accuracy across three different settings as discussed in subsection 5.1. FGC, FGCL, and L.Data utilize the experimental settings discussed in subsection 5.1 of the experimental setup, specifically Set3(a), Set3(b), and Set1, respectively. The results clearly demonstrate that training on a learned graph significantly increases testing accuracy. However, it is also evident that learning a graph for the Coauthor Physics and Coauthor CS datasets requires substantial memory, resulting in out-of-memory(OOM) errors on x86-64 processor having 48CPU and 96GB RAM. Additionally, learning a coarsened graph using proposed SCGL and training the GNN with this coarsened graph substantially improves testing accuracy compared to existing state-of-the-art methods.

## 5.3 TIME COMPLEXITY

The per-iteration time complexity of the proposed CGL and SCGL algorithms is $\mathcal{O}(kpn + k^2p)$, where $p$ represents the number of original data points, $k$ is the number of nodes in the coarsened graph, and $n$ is the dimension of each data point $\mathbf{x}_i$. In comparison, the time complexity of existing state-of-the-art graph coarsening techniques varies significantly. The GCOND algorithm, for instance, has a time complexity of $\mathcal{O}(k^2d^2 + r^Lpd^2 + Lk^2d + Lkd)$, where $r$ is the number of sampled neighbors per node, $L$ is the number of MLP layers, and $d$ is the number of hidden units for all layers. On the other hand, the FGC algorithm has a time complexity of $\mathcal{O}(7p^2k + 5k^2p + 5kpn + k^3)$, with $p$ representing the number of features, $k$ the number of clusters, and $n$ the number of nodes. It is evident from Table 6 that proposed CGL algorithm is much faster than existing state of the art graph coarsening techniques.

## 5.4 COARSENING UNDER ADVERSARIAL CONDITIONS

Graph Neural Networks (GNNs) are known to exhibit instability under perturbations in input data, with their performance heavily reliant on the training graph. This sensitivity makes them particularly vulnerable to structural changes and minor perturbations [38; 39; 40]. Adversarial attacks, which involve malicious modifications to the graph structure or features, can significantly degrade the performance of GNNs by causing incorrect predictions [24; 41; 42; 43].

Coarsening on a poisoned dataset further exacerbates this instability, making it challenging to maintain robust performance. Our proposed framework mitigates the impact of adversarial attacks while simultaneously learning coarsened structure and feature matrices. Notably, since our framework

| Dataset/$\tau$(sec.) | $\tau_l$ | r | $\tau_{FGC}$ | $\tau_{Total}$ | $\tau_{CGL}$ |
|---|---|---|---|---|---|
| Cora | 114.23 | 0.3 | 12.95 | 127.18 | **5.43** |
| | | 0.1 | 5.86 | 120.09 | **1.19** |
| | | 0.05 | 4.84 | 119.07 | **0.41** |
| Citeseer | 170.55 | 0.3 | 17.67 | 188.22 | **6.82** |
| | | 0.1 | 9.70 | 180.25 | **2.14** |
| | | 0.05 | 7.07 | 177.62 | **0.64** |
| Pubmed | 10162.18 | 0.05 | 73.42 | 10235.60 | **8.69** |
| | | 0.03 | 46.66 | 10208.24 | **5.55** |
| | | 0.01 | 26.12 | 10188.3 | **1.58** |
| Co-phy | OOM | 0.05 | 2950.25 | OOM | **36.10** |
| | | 0.03 | 1640.79 | | **15.16** |
| | | 0.01 | 338.90 | | **5.63** |
| Co-cs | OOM | 0.05 | 1177.32 | OOM | **10.44** |
| | | 0.03 | 622.61 | | **6.30** |
| | | 0.01 | 66.92 | | **1.63** |

Table 6: This table compares the time required for graph learning and coarsening processes. Specifically, $\tau_l$ represents the time to learn the graph from data $X$ [6], $\tau_{FGC}$ is the time to learn a coarsened graph using the FGC method [13], and $\tau_{Total}$ is the total time required to learn and coarsen the graph using FGC. Additionally, $\tau_{CGL}$ denotes the time required to learn the coarsened graph directly from raw data $X$ using our proposed CGL method. Note that for the Coauthor Physics and Coauthor CS datasets, a out of memory error(OOM) occurred on a x86-64 processor having 48CPU and 96GB RAM. For these datasets, we reported $\tau_{FGC}$ considering the original graph Laplacian matrix. It is evident that the proposed CGL algorithm is significantly faster than the existing state-of-the-art methods.

| pr (%) | Cora | Citeseer | Co-phy | Co-cs | Pubmed |
|---|---|---|---|---|---|
| 15 | 76.77 | 70.39 | 94.93 | 90.09 | 84.00 |
| 10 | 80.76 | 70.63 | 95.01 | 90.53 | 84.59 |
| 5 | 78.43 | 70.99 | 94.94 | 89.82 | 85.34 |
| 0 | 79.02 | 70.60 | 94.76 | 88.69 | 84.73 |

Table 7: Node classification performance under an adversarial attack on the graph's feature matrix. The table demonstrates that increasing the perturbation rate minimally impacts performance.

is designed without direct reliance on structure information, it remains robust to perturbations in the structure matrix. Thus, structural perturbations introduced during coarsening do not degrade its performance. For evaluation, we focus on feature attacks, where a random subset of elements in the feature matrix $X$ is perturbed by setting them to zero. We define the *perturbation rate (pr)* as the percentage of perturbed elements relative to the total number of elements. Table 7 presents the performance of CGL algorithms on node classification tasks under various perturbation rates. The results demonstrate the resilience of our approach to feature perturbations and its effectiveness in maintaining stable performance across adversarially attacked graphs.

## 6 CONCLUSION

This paper introduces a novel Coarsened Graph Learning (CGL) framework that directly learns a coarsened graph from raw data or features, eliminating the need for a graph structure. The framework also includes a pipeline for GNNs that utilizes only feature data. By incorporating label information, CGL enhances performance in node classification tasks. Extensive experiments show that CGL outperforms existing state-of-the-art methods in terms of accuracy and efficiency. Additionally, CGL demonstrates robustness against adversarial attacks and generalizability across various GNN architectures, maintaining high node classification performance.

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

# A    APPENDIX

## A.1    PROOF OF THEOREM 1

We show that each limit point $(\mathbf{w}^t, P^t, \tilde{X}^t)$ satisfies KKT condition for equation 9. Let $(\mathbf{w}^\infty, P^\infty, \tilde{X}^\infty)$ be a limit point of the generated sequence. Next, the Lagrangian function of equation 9 is:

$$L(\mathbf{w}, P, \tilde{X}, \boldsymbol{\mu}_1, \boldsymbol{\mu}_2) = -\gamma \log \det(\mathcal{L}\mathbf{w} + J) + \operatorname{tr}(\tilde{X}^T \mathcal{L}\mathbf{w}\tilde{X}) + \frac{\delta}{4}\|PP^T - I\|_F^2 - \boldsymbol{\mu}_1^\top \mathbf{w}$$

$$+ \frac{\alpha}{2}\|\tilde{X} - PX\|_F^2 + \frac{\beta}{2}\|\mathcal{L}\mathbf{w}\|_F^2 - \boldsymbol{\mu}_2^\top P + \boldsymbol{\mu}_3^T \Big[ \|P_1^T\|_2^2 - 1 \quad \|P_2^T\|_2^2 - 1 \ldots \|P_k^T\|_2^2 - 1 \Big]^T \quad (21)$$

where $\boldsymbol{\mu}_1$, $\boldsymbol{\mu}_2$ and $\boldsymbol{\mu}_3$ are the dual variables.
(1) The KKT condition with respect to $\mathbf{w}$ is:

$$-\gamma \mathcal{L}^*(\mathcal{L}\mathbf{w}^t + J)^{-1} + \beta \mathcal{L}^*(\mathcal{L}\mathbf{w}^t) + \mathcal{L}^*(\tilde{X}\tilde{X}^T) - \boldsymbol{\mu}_1 = 0 \quad (22)$$

$$\boldsymbol{\mu}_1^\top \mathbf{w} = 0, \quad (23)$$

$$\boldsymbol{\mu}_1 \geq 0, \quad (24)$$

$$\mathbf{w} \geq 0, \quad (25)$$

w is derived by using KKT condition from equation 13:

$$\mathbf{w}^\infty - \mathbf{w}^\infty + \frac{1}{L}\Big( -\frac{\gamma}{\beta}\mathcal{L}^*(\mathcal{L}\mathbf{w}^\infty + J)^{-1} + \mathcal{L}^*(\mathcal{L}\mathbf{w}^\infty) + \mathcal{L}^*(\frac{1}{\beta}\tilde{X}\tilde{X}^T)\Big) = 0 \qquad (26)$$

For $\boldsymbol{\mu}_1 = 0$, we observe that $\mathbf{w}^\infty$ satisfies the KKT condition.
(2) The KKT condition with respect to $P$ is

$$-\alpha(\tilde{X} - PX)X^T + \delta(PP^T - I)P - \boldsymbol{\mu}_2 + 2\Big[\mu_{31}P_1^T, \ldots \mu_{3k}P_k^T\Big]^T = 0 \qquad (27)$$

$$\boldsymbol{\mu}_3^T\Big[\|P_1^T\|_2^2 - 1 \quad \|P_2^T\|_2^2 - 1 \ldots \|P_k^T\|_2^2 - 1\Big]^T = 0, \qquad (28)$$

$$\boldsymbol{\mu}_2^\top P = 0, \qquad (29)$$

$$\boldsymbol{\mu}_2 \geq 0, \qquad (30)$$

$$P \geq 0, \qquad (31)$$

$$\boldsymbol{\mu}_3 \geq 0, \qquad (32)$$

$$\|[P^T]_i\|_2^2 \leq 1 \qquad (33)$$

For $\boldsymbol{\mu}_2 = \phi$ and $\mu_{2i}[P^T]_i^\infty = 0 \ \forall\ i = 1, 2, \ldots k$, we observe that $P^\infty$ satisfies the KKT condition from 17.
(3) The KKT condition with respect to $\tilde{X}$ is

$$2\mathcal{L}\mathbf{w}\tilde{X} + \alpha(\tilde{X} - PX) = 0$$

This concludes the proof.

## A.2 Datasets

| Dataset | Nodes (p) | Edges (e) | Features (n) | Classes (c) |
|---|---|---|---|---|
| Cora | 2708 | 5429 | 1433 | 7 |
| Citeseer | 3327 | 9104 | 3703 | 6 |
| Pubmed | 19717 | 44338 | 500 | 3 |
| Coauthor CS | 18333 | 163788 | 6805 | 15 |
| Coauthor Physics | 34493 | 247962 | 8415 | 5 |
| Cornell | 183 | 295 | 1703 | 5 |
| Texas | 183 | 309 | 1703 | 5 |
| Wisconsin | 251 | 499 | 1703 | 5 |
| Chameleon | 2277 | 31421 | 2325 | 5 |
| Squirrel | 5201 | 198493 | 2089 | 5 |
| Flickr | 89250 | 899756 | 500 | 7 |
| Genius | 421961 | 922868 | 12 | 2 |
| Arxiv-year | 169343 | 1166243 | 128 | 5 |
| OGBN-Proteins | 132534 | 39561252 | 8 | 112 |
| OGBN-Product | 2449029 | 61859140 | 100 | 47 |

Table 8: Node Classification Datasets

The hyperparameters for graph coarsening and GNN model are tuned using grid search. The learning rate and decay rate used in the node classification experiments are 0.01 and 0.0001 respectively.

## A.3 Coarsening of Heterophily Datasets

We present the coarsening results on different heterophily datasets in the Table 9.

## A.4 Generalizability of Proposed CGL Algorithm

Next, in this subsection, we will illustrate the generalizability of the learned coarsened graph from the proposed CGL algorithm by using different architectures to train the GNN. Specifically, we have used GNN architectures like GCN [44], APPNP [45], and GAT [46] to train our GNN and perform the node classification task. Table 10 demonstrates that CGL is compatible with different widely used GNN architectures, giving almost similar Node Classification accuracy across all the datasets.

| Dataset | r | GCOND | FGC | CGL | SCGL | Whole Dataset |
|---------|---|-------|-----|-----|------|---------------|
| Texas | 0.7 | - | 69.74 | 68.3 | **71.18** | |
| | 0.5 | - | 67.97 | 68.85 | **70.49** | 56.18 |
| | 0.3 | - | 65.02 | 66.12 | **66.67** | |
| Cornell | 0.7 | 51.25 | 62.86 | 61.25 | **63.93** | |
| | 0.5 | 45.94 | 60.80 | 61.26 | **62.29** | 52.10 |
| | 0.3 | 43.24 | 58.25 | 60.11 | **61.75** | |
| Chameleon | 0.7 | 38.78 | 48.25 | 57.4 | **59.33** | |
| | 0.5 | 36.25 | 46.96 | 53.35 | **56.17** | 39.38 |
| | 0.3 | 37.7 | 37.74 | 50.50 | **53.84** | |
| Squirrel | 0.7 | - | 34.83 | 45.59 | **46.72** | |
| | 0.5 | - | 33.29 | 42.05 | **42.99** | 24.3 |
| | 0.3 | - | 28.74 | 36.72 | **37.75** | |
| Wisconsin | 0.7 | 58.8 | 60.68 | 61.35 | **62.15** | |
| | 0.5 | 56.32 | 60.78 | 60.95 | **61.75** | 66.9 |
| | 0.3 | 54.24 | 58.88 | 60.15 | **60.56** | |

Table 9: This table shows the node classification performance on the heterophilic datasets. It is evident that for coarsening, we have utilized only the feature matrix $X$ of the graph, and yet our framework outperforms or comparable against the existing state-of-the-art graph coarsening methods, which consider both the Laplacian and the feature matrix for coarsening. Additionally, according to the available code, SCAL shows errors on heterophilic datasets, and GCOND shows errors on some datasets, such as Texas and Squirrel.

| Dataset | GCN | APPNP | GAT |
|---------|-----|-------|-----|
| Cora | 79.025 | 79.57 | 77.32 |
| Citeseer | 70.60 | 70.42 | 68.29 |
| Pubmed | 84.73 | 84.49 | 84.25 |
| Co-cs | 88.69 | 86.40 | 87.67 |

Table 10: This table depicts the CGL Node Classification using multiple GNN architectures for a coarsening ratio of 0.05. It is evident that the proposed CGL algorithm can be utilized with any GNN architecture without compromising accuracy.

| Dataset | r | Proposed SCGL (Coarsening+ Classification) | Whole Dataset |
|---------|---|------------------------------------------|---------------|
| Cora | 0.1 | **556** | 4096 |
| Citeseer | 0.05 | **597** | 8099 |
| Flickr | 0.01 | **1126** | 86016 |
| OGBN-Proteins | 0.001 | **1638** | 91136 |
| Arxiv-year | 0.001 | **978** | 95232 |

Table 11: The table indicates that graph coarsening significantly reduces **memory usage (in MB)** for node classification tasks compared to using the original dataset.

