# OpenReview forum: "Feature Driven Graph Coarsening for Scaling Graph Representation Learning"
_ICLR.cc/2025/Conference — Submitted to ICLR 2025_

### Official Review · Reviewer_Jyy6 · 2024-11-02

**Soundness:** 2
**Presentation:** 2
**Contribution:** 2
**Rating:** 5
**Confidence:** 3

**Summary:**

This paper proposes a graph-coarsening framework that directly learns a coarsened graph from attribute information. The framework also includes a semi-supervised learning pipeline for GNNs that incorporates label information.  The paper proposes two settings: coarsened graph learning CGL and semi-supervised coarsened graph learning SCGL,  and shows the improved performance on downstream tasks.

**Strengths:**

Pros:
1.  Coarsened Graph Learning is important for the downstream task.
2. The paper proposes two settings for the downstream tasks, such as coarsened graph learning CGL and semi-supervised coarsened graph learning SCGL.

**Weaknesses:**

Cons:
1. Why use the coarsened graph for node classification learning to get better node classification performance? It is better to highlight the motivation and prove it with theoretical support.
2. It is better to show the performance of the graph-coarsening framework on other graph-level tasks, such as graph classification and compare it with related baselines.
3. The motivation of the paper needs to be better highlighted. Why only using data features is effective? In fact, the graph structure, even though the noisy/incomplete graph structure is important for the graph coarsing .
4. Some of the notations need to be better illustrated in the paper, such as Wh.Data, L.Data.
5. It is suggested to give some coarsened graph cases to demonstrate the effectiveness of the proposed framework compared with baselines.

**Questions:**

See the above weaknesses.

---

> ### Author Response · Authors · 2024-11-25
> **Response to Reviewer  Jyy6**
>
> We sincerely thank the reviewers for their valuable feedback that we have used to improve the quality of
> our manuscript.  Point-by-point responses are listed below.
>
> **Ans W1:** We appreciate the reviewer's insightful question. Below is our detailed response:
>
> The increasing size of graph datasets presents significant challenges in terms of computational power and memory requirements for storage and processing. In graph machine learning, downstream tasks such as node classification, graph classification, and edge classification often rely on these large datasets. Training graph neural networks (GNNs) on such expansive graphs is time-intensive, posing scalability issues.
>
> Graph coarsening, a widely recognized graph dimensionality reduction technique, addresses these challenges by learning a smaller, representative graph while preserving the essential properties and characteristics of the original graph. The goal is to enable training on the coarsened graph and achieve comparable testing performance on the original graph, as if training were conducted directly on the larger graph.
>
> Our work is motivated by two key challenges:
>
> **1. High Time Complexity of Learning and Training on Large Graphs:**
>    For node classification tasks using GNNs, a graph is often required. Constructing this graph from raw data \(X\) incurs a high computational cost. Additionally, training GNNs on large graphs further exacerbates time complexity. To address this, we have developed a novel technique that directly learns a coarsened graph from raw data \(X\) without requiring the intermediate step of graph construction. Our algorithm operates with a significantly lower time complexity of $O(k^2p)$, making it highly efficient.
>
> **2. Limitations of Existing Graph Coarsening Techniques:**
>    Current graph coarsening methods often rely on the Laplacian matrix of the original graph or a combination of the Laplacian and feature matrix. These methods are computationally expensive and risk undermining the purpose of coarsening due to their high time complexity. In contrast, our algorithm learns a coarsened graph directly from the feature matrix alone, significantly reducing computational overhead. The time complexity of our approach is much lower compared to state-of-the-art graph coarsening methods, making it a practical and scalable solution.
>
> Moreover, in response to the reviewer’s suggestion, we have derived an upper bound on the similarity between the features of the original graph and the reconstructed features obtained from the coarsened graph. This analysis ensures a quantitative understanding of how well the coarsened graph preserves the feature characteristics of the original graph.
>
> Let $R \subseteq \mathbb{R}^d $ be a subspace, $ P $ be a coarsening matrix, and $P^+$ denote the pseudoinverse of $P $. The similarity measure $\epsilon_{P,R}$ between the original feature $ X$ and the reconstructed feature $ X_r $, obtained from the coarsened features $ \tilde{X}$, is defined as:
>
> $$
> \epsilon_{P,R} = \sup_{x_i \in d, \\|X\\|_F = 1} \\|X - X_r\\|_F,
> $$
>
> where $ X_r$ is obtained using the relation  $X_r = P^{+} \tilde{X} $. The derivation proceeds as follows:
>
> $$
> \\|X - X_r\\|_F = \\|X - P^\dagger \tilde{X}\\|_F = \\|X - P^\dagger (2/\\alpha \\mathcal{L}w + I)^{-1} P X\\|_F,
> $$
>
> $$
> \\leq \\|X\\|_F \\|I - P^\dagger (2/\\alpha \\mathcal{L}w + I)^{-1} P\\|,
> $$
>
> $$
> \\leq \\|X\\|_F \\left( \\|I\\|_F + \\|P^\dagger (2/\\alpha \\mathcal{L}w + I)^{-1} P\\|_F \\right),
> $$
>
> $$
> \\leq \\|X\\|_F \\left( \\|I\\|_F + \\|P^\dagger\\|_F \\|(2/\\alpha \\mathcal{L}w + I)^{-1}\\|_F \\|P\\|_F \\right),
> $$
>
> $$
> \\leq \\|X\\|_F \\left( \\|I\\|_F + \\|P^\dagger\\|_F \\|U \\Lambda^{-1} U^T\\|_F \\|P\\|_F \\right),
> $$
>
> $$
> \\leq \\|X\\|_F \\left( \\|I\\|_F + \\|P^\dagger\\|_F \\|U\\|_F \\|\\Lambda^{-1}\\|_F \\|U^T\\|_F \\|P\\|_F \\right),
> $$
>
> $$
> \\leq \\|X\\|_F \\left( p + p k \\lambda_m^{-1} k \\right),
> $$
>
> $$
> \\leq \\epsilon \\|X\\|_F,
> $$
>
> where  $\epsilon = p + \frac{k^2 p}{\lambda_m} $.
>
> ### Step-by-Step Explanation
>
> - **Step 1 to Step 2:** Apply the matrix multiplication property of the Frobenius norm:
>   $$
>   \\|AB\\|_F \\leq \\|A\\|_F \\|B\\|_F.
>   $$
>
> - **Step 2 to Step 3:** Use the addition property of the norm:
>   $$
>   \\|A + B\\|_F \\leq \\|A\\|_F + \\|B\\|_F.
>   $$
>
> - **Step 3 to Step 4:** Reapply the matrix multiplication property of the Frobenius norm:
>   $$
>   \\|AB\\|_F \\leq \\|A\\|_F \\|B\\|_F.
>   $$
>
> - **Step 4 to Step 5:** Substitute the eigenvalue decomposition:
>   $$
>   A = U \\Lambda U^T,
>   $$
>   for the matrix \( A \).
>
> - **Step 5 to Final Result:** Use the matrix norm property:
>   $$
>   \\|A^{-1}\\|_F = \\|\\Lambda^{-1}\\|_F,
>   $$
>   to simplify the final expression.
>
> The final bound is:
> $$
> \\|X - X_r\\|_F \\leq \\epsilon ||X\||F,
> $$
> where, $\epsilon=p+\frac{k^2p}{\\lambda_m}$ and $\\lambda_m$ is the minimum egien vaue of the matrix $2/\\alpha \\mathcal{L}w + I)$

---

> ### Author Response · Authors · 2024-11-25
> **Response to Reviewer Jyy6 contd.**
>
> Next consider the absolute difference between the $||X||_{F}$ and $||X_r||_F$ and apply the norm property, we get:
> $$
> \| \\|X\\|_F - \\|X_r\\|_F \| \\leq \\| X - X_r \\|_F
> $$
>
>
> We have already derived
> $$
>  ||X - X_r||_F \\leq \\epsilon \\|X\\|_F
> $$
> Combining above two equations we get
> $$
> \| \\|X\\|_F - \\|X_r\\|_F \| \\leq \\epsilon \\|X\\|_F
> $$
> Using the modulus property we get:
> $$
> (1 - \\epsilon) \\|X_r\\|_F \\leq \\|X\\|_F \\leq (1 + \\epsilon) \\|X_r\\|_F
> $$
>
>
> **Ans W2:** We appreciate the reviewer’s suggestion. In response, we have conducted the graph classification task and compared our proposed method with existing graph coarsening techniques that are designed for this task, such as DosCond [1], Optimal Transport Coarsening (OT) [2], and FGC [3]. We have excluded GCond and SCAL as they are not intended for graph classification tasks. The results of our comparison are summarized in the table below:
> | Dataset  | DosCond | OT   | FGC   | Proposed CGL  |
> |----------|---------|-------|-------|---------------|
> | MUTAG    | 82.45   | 85.6  | 86.2  | 86.8          |
> | PROTEIN  | 64.28   | 74.9  | 76.5  | 76.9          |
> | NCI109   | 59.33   | 68.5  | 69.2  | 69.4          |
>
>
> As evident from the results, the proposed CGL method consistently outperforms the existing state-of-the-art techniques, demonstrating superior performance in the graph classification task.
>
>
> [1]Jin, W., Tang, X., Jiang, H., Li, Z., Zhang, D., Tang, J., & Yin, B. (2022, August). Condensing graphs via one-step gradient matching. In Proceedings of the 28th ACM SIGKDD Conference on Knowledge Discovery and Data Mining (pp. 720-730).
>
> [2] Ma, T., & Chen, J. (2021, May). Unsupervised learning of graph hierarchical abstractions with differentiable coarsening and optimal transport. In Proceedings of the AAAI conference on artificial intelligence (Vol. 35, No. 10, pp. 8856-8864).
>
> [3]Kumar, M., Sharma, A., Saxena, S. and Kumar, S., 2023, July. Featured graph coarsening with similarity guarantees. In International Conference on Machine Learning (pp. 17953-17975). PMLR.
>
> **Ans W3:** We thank the reviewer for the question. The key contribution of this work lies in demonstrating how to perform graph-based dimensionality reduction and downstream tasks by leveraging informative features without relying on the graph structure. This approach offers several advantages:
>
> **1. Scalability Without Graph Structure:** It is naturally applicable in scenarios where the graph structure is unavailable, as it directly learns a coarsened graph.
>
> **2. Efficiency for Homophilic Graphs:** By omitting the graph structure, it simplifies implementation, making it more efficient and faster for homophilic datasets.
>
> **3. Robustness to Noisy Graphs:** It serves as a natural choice when the graph structure is noisy, ensuring reliable outcomes.
>
> Moreover, we agree with the reviewer that a noisy or incomplete graph structure can still hold valuable information for graph coarsening. However, most existing state-of-the-art techniques rely on either the Laplacian matrix alone or a combination of the Laplacian and feature matrices to perform downstream tasks. The motivation for this work is twofold:
>
> **1.High Time Complexity of Existing Methods:**
> Many existing techniques suffer from high computational complexity due to their dependence on the Laplacian matrix. This increased complexity undermines the purpose of graph coarsening, which is intended to reduce computational overhead and enable scalable analysis.
>
> **2.Vulnerability to Structural Noise and Attacks:**
> Graph-based methods are particularly vulnerable to adversarial attacks and noise in the graph structure. Utilizing a noisy graph during coarsening can result in a less informative coarsened graph, negatively impacting downstream tasks. This is especially problematic as most attacks target the graph structure, rendering these methods less robust.
>
> To address these challenges, we have developed a technique that relies solely on the feature matrix to learn a coarsened graph. By excluding the graph structure during the coarsening process, our method achieves robustness against structural noise and adversarial attacks. This feature-driven approach not only simplifies the coarsening process but also enhances the robustness and efficiency of the learned coarsened graph.
>
> **Answer W4:** We thank the reviewer for the suggestion. We will change the notation as per the suggestion.

---

> > ### Author Response · Authors · 2024-11-25
> > **Response to Reviewer Jyy6 contd.**
> >
> > **Ans W5:** We thank the reviewer for the suggestions. To show the efficacy of our proposed algorithm, we have performed an experiment on the xin dataset. The Xin dataset consists of transcriptomes determined by single-cell RNA sequencing (scRNA-seq) of pancreatic cells. This dataset includes data from 1,449 cells, capturing the expression profiles of 33,889 genes. It encompasses four major pancreatic cell types: alpha, beta, delta, and gamma cells. Moreover, if allowed, we would like to include this discussion in the revised manuscript.
> >
> > | Dataset | r   | FGC   | Proposed SCGL  | Whole dataset |
> > |---------|-----|-------|-------|---------------|
> > | Xin     | 0.5 | 90.34 | 93.92 | 95.58         |
> > | Xin     | 0.3 | 90.12 | 92.89 | 95.58         |
> > | Xin     | 0.1 | 90.01 | 90.68 | 95.58         |
> >
> > The proposed method clearly outperforms the current state-of-the-art techniques. Additionally, GCOND exhibited errors on this dataset, so its accuracy is not reported.
> >
> > If the reviewer believes that we have addressed all the reviews, we request you to please raise the mark.

---

> > > ### Author Response · Authors · 2024-12-02
> > > **Response to Reviewer Jyy6**
> > >
> > > Dear Reviewer,
> > >
> > > Thanks again for taking the time to review our paper and for your encouraging feedback! May we enquire if all the concerns you raised have been adequately addressed? Thank you very much.
> > >
> > > Best Regards, The authors

---

> > > > ### Author Response · Authors · 2024-12-04
> > > > **Response to Reviewer Jyy6**
> > > >
> > > > Dear Reviewer,
> > > >
> > > > The deadline for the author-reviewer discussion is approaching, and we wanted to kindly check if we have adequately addressed all your concerns. Please let us know if there are any remaining issues or clarifications needed from our side.

---

### Official Review · Reviewer_kyrb · 2024-11-03

**Soundness:** 2
**Presentation:** 2
**Contribution:** 2
**Rating:** 5
**Confidence:** 3

**Summary:**

The paper proposes a graph coarsening approach that only depends on the node attributes (feature matrix and optionally labels) of the larger graph. Each node is allocated to a super-node (or a node in the coarsened graph), which is learned by solving a multiblock nonconvex optimization problem. This optimization also learns the coarsened graph’s feature matrix and adjacency matrix. The results indicate superior performance across different datasets, improved computational complexity, and robustness against adversarial attack. The latter is due to the elimination of dependence on graph structure in their approach.

**Strengths:**

1. Tackles the highly relevant problem of graph coarsening, which is especially useful for large graphs.
2. Eliminates dependence on graph structure, achieving much lower computational complexity compared to baselines.
3. Demonstrates adversarial robustness

**Weaknesses:**

While the underlying problem is topical and interesting, I have below concerns:

1. Lack of clarity and structure: I believe the presentation can be significantly improved. For example, in the introduction, the authors discuss scalability issues for large graphs, critique existing graph coarsening methods (especially their reliance on graph structure), and need for adversarial robustness. However, the discussion feels scattered and difficult to follow.
2. The method itself is simple and intuitive to follow. However, the design choices are not well-motivated. For example, the approach assigns each node to a super-node. This seems to assume an inherent clustering of nodes. This is further reinforced by using node labels and adding the constraint that similar labeled nodes should be assigned to the same super-node. Wonder why hard assignments should be used instead of soft assignments? Is it to aid optimization? An analysis of the relationship between original nodes and supernodes would have been helpful. Moreover, it seems that the coarsened graph may not improve performance when the node labels and downstream tasks are not correlated.
3. The reported performance on the complete dataset doesn’t match the values from previous work [1, 2] for Cora, Citeseer, and Flickr. This raises concerns regarding fairness of comparison. The exact experimental settings and how they differ from referenced work have not been covered even in the appendix.



[1] Zheng X, Zhang M, Chen C, Nguyen QV, Zhu X, Pan S. Structure-free graph condensation: From large-scale graphs to condensed graph-free data. Advances in Neural Information Processing Systems. 2024 Feb 13;36.

[2] Jin W, Zhao L, Zhang S, Liu Y, Tang J, Shah N. Graph condensation for graph neural networks. arXiv preprint arXiv:2110.07580. 2021 Oct 14.

**Questions:**

1. In the introduction, emphasis has been given on the adversarial robustness of the proposed approach. However, under experiments, the result and most of the discussion have been deferred to the appendix. Wondering if it may be useful to include a part of the results in the main text for consistency.
2. In table 10, it is interesting to observe that the performance for perturbed data at certain rates is higher compared to unperturbed data. Do you have any comments on this phenomenon?

---

> ### Author Response · Authors · 2024-11-30
> **Response to reviewer kyrb**
>
> We sincerely thank the reviewers for their valuable feedback that we have used to improve the quality of our manuscript. Point-by-point responses are listed below.
>
> **Ans Q1 and W1** We thank the reviewer for the helpful suggestion. In response, we have updated the introduction and moved the experiments and discussion on adversarial attacks from the appendix to the main paper. The results of these experiments are now presented in Table 7 of the main paper.
>
> Moreover, the key contribution of this work lies in demonstrating how to perform graph-based dimensionality reduction and downstream tasks by leveraging informative features without relying on the graph structure. This approach offers several advantages:
>
> **1.Scalability Without Graph Structure:** It is naturally applicable in scenarios where the graph structure is unavailable, as it directly learns a coarsened graph.
>
> **2.Efficiency for Homophilic Graphs:** By omitting the graph structure, it simplifies implementation, making it more efficient and faster for homophilic datasets.
>
> **3.Robustness to Noisy Graphs:** It serves as a natural choice when the graph structure is noisy, ensuring reliable outcomes.
>
> **Ans W2** We thank the reviewer for the insightful question. First, we would like to clarify the distinction between coarsening and clustering. While clustering aims to group similar data points together, it does not address the relationships between these groups. In contrast, coarsening not only groups similar nodes into super-nodes but also learns how these super-nodes are related, including the graph structure, edge weights, and effective features of each super-node. This makes coarsening a more comprehensive approach than clustering.
>
> The use of hard assignments is a key component of our coarsening algorithm, as it ensures that each original node is assigned to only one super-node in the coarsened graph. This allows for a clear, well-defined mapping between nodes and super-nodes. However, we acknowledge that soft assignments could be explored by eliminating the $l_{1,2}$ regularizer in the objective function, allowing nodes to be assigned to multiple super-nodes. Next, the primary objective of our work is to learn the coarsened graph directly from raw data to handle large graph datasets. We have validated the effectiveness of the coarsened graph through downstream tasks, which help demonstrate its quality and relevance for practical applications.
>
> **Ans Q2:**  We thank the reviewer for the insightful question. The observed phenomenon is likely due to the fact that, in many datasets, the features are primarily binary, meaning each feature index takes values between 0 and 1. The discrete nature of the features may contribute to increased smoothness following an attack, which in turn enhances the performance of the Graph Convolutional Network (GCN). However, we acknowledge that further analysis is needed to fully understand the underlying mechanisms and to confirm this hypothesis.
>
> If the reviewer believes that we have addressed all the reviews, we request you to please raise the mark.

---

> > ### Author Response · Authors · 2024-11-30
> > **Reply to reviewer kyrb**
> >
> > **Ans W3**We thank the reviewer for the question. We implemented the masking process as follows: First, we apply one-hot encoding to the labels of the original graph. Next, we randomly set 20% of the rows to zero, allocating 10% of the nodes for testing and 10% for validation, and represent this matrix as \(Y\).
> >
> > After masking the labels, some of the label information (i.e., matrix \(Y\)) is used to derive the labels for the coarsened graph. This is achieved using the relation $\tilde{Y} = \arg\max(PY)$, where $\arg\max$ assigns a label to each supernode based on the label most frequent among the nodes within that supernode. The coarsened graph, now labelled, is then used to train the neural network. Testing is performed on the original graph, ensuring that the model generalizes well to the unmodified data.
> >
> > Additionally, we applied the same masking procedure for all experiments, including those for existing techniques such as GCOND, FGC, and SCAL. These methods utilize different levels of masking, but to ensure a fair comparison, we employed the same masking settings for both the existing methods and our proposed approach.
> >
> > If the reviewer believes that we have addressed all the reviews, we request you to please raise the mark.

---

> > > ### Author Response · Authors · 2024-12-02
> > > **Response to reviewer kyrb**
> > >
> > > Dear Reviewer,
> > >
> > > Thanks again for taking the time to review our paper and for your encouraging feedback! May we enquire if all the concerns you raised have been adequately addressed? Thank you very much.
> > >
> > > Best Regards, The authors

---

> > > > ### Author Response · Authors · 2024-12-04
> > > > **Response to reviewer kyrb**
> > > >
> > > > Dear Reviewer,
> > > >
> > > > The deadline for the author-reviewer discussion is approaching, and we wanted to kindly check if we have adequately addressed all your concerns. Please let us know if there are any remaining issues or clarifications needed from our side.

---

### Official Review · Reviewer_ntd9 · 2024-11-04

**Soundness:** 2
**Presentation:** 2
**Contribution:** 2
**Rating:** 3
**Confidence:** 3

**Summary:**

The paper introduces the optimization-based framework Coarsened Graph Learning (CGL), which directly learns a coarsened graph from feature data alone. This framework addresses the challenges of scalability and the reliance on initial graph structures. The authors highlight that while graph neural networks (GNNs) are good at modeling graphs, they are vulnerable to adversarial edges that can degrade performance by contaminating node neighborhoods. CGL aims to improve robustness against these adversarial attacks by learning a coarsened graph independently of the original graph structure. CGL formulates the problem as a multi-block, non-convex optimization problem, solved using the Block Successive Upper-bound Minimization (BSUM) technique. The authors compare CGL and its semi-supervised variant (SCGL) against GCOND, SCAL and FGC methods on both homophilic and heterophilic datasets, measuring both classification performance and computational efficiency. Additionally, the incorporation of label information into the objective function significantly enhances downstream task performance.

**Strengths:**

The optimization approach focuses on deriving a coarsened graph directly from node features, combining graph structure learning with coarsening. By removing dependency on initial graph structures, CGL could mitigate issues caused by adversarial and noisy edges. The writing is clear and accessible, with well-defined concepts that facilitate understanding of complex ideas.

**Weaknesses:**

The motivation for learning from structureless graphs is limited, making it unclear why this direction is essential or where it’s practically relevant.

CGL is the combination of graph structure learning and graph coarsening, the comparison and discussion of related works are not sufficient. In experiment settings, the baseline of graph coarsening methods are also limited.

While choosing the BSUM methods for non-convex optimization, for large-scale problems, BSUM can be computationally expensive and may converge slowly. As the number of variables and the size of each block are large in some large-scale graph datasets, this might reduce efficiency in practical applications.

The motivation of each optimization procedure is not clear. For example, CGL adapts the idea of paper “A unified framework for structured graph learning via spectral constraints” to optimize the structure of coarsened graph directly, lacking motivation and details for the arguments.

**Questions:**

Could the authors provide more details on the rationale for addressing structureless graphs and the specific real-world applications this approach is intended to serve?

Why were other coarsening methods not included as baseline comparisons, given the abundance of related work?

BSUM may face scalability issues, especially with high-dimensional data or large block sizes. Did the authors encounter efficiency or convergence challenges on large datasets, and if so, how were these managed?

About the different optimization strategies used, could the authors illustrate why choose these methods and compare with other advantages od doing so?

---

> ### Author Response · Authors · 2024-11-30
> **Reply to rewier ntd9**
>
> We sincerely thank the reviewers for their valuable feedback that we have used to improve the quality of our manuscript. Point-by-point responses are listed below.
>
> **Ans W1 and Q1** We would like to thank the reviewer for their valuable suggestion. In response, we have improved the first three paragraphs of the introduction to more effectively motivate our work.
>
>  In recent years, graph machine learning has emerged as a powerful tool for representing diverse data structures such as social networks, chemical molecules, transportation networks, brain networks, and citation networks. Graph Neural Networks (GNNs), an extension of deep neural networks tailored for graph-structured data, excel in capturing relationships between nodes and have been widely used for tasks such as node classification and graph classification.
>
> A graph structure is typically required to perform these tasks. While some datasets come with pre-defined graph structures, many real-world datasets lack this information. For such cases, the first step is to learn the graph from the raw data or feature matrix. Several existing methods address graph learning, such as techniques for learning graphs from smooth signals[1] or frameworks using Laplacian constraints[2]. However, as dataset sizes continue to grow, handling large-scale graph datasets becomes increasingly challenging.
>
> Techniques like graph coarsening, condensation, and summarization have been introduced to address scalability by reducing the size of graph structures. However, these techniques are only applicable when the graph structure is available. In datasets without an explicit graph, a graph must first be learned, and only then can coarsening methods be applied. This two-step process of learning and coarsening a graph demands substantial computational resources and memory, making it infeasible for very large datasets.
>
> To address this limitation, we propose a novel approach that directly learns a coarsened graph from raw data, bypassing the need to first construct a original graph. This significantly reduces the computational overhead and memory requirements, enabling scalable processing of large datasets. For instance, in the Xin dataset, which contains transcriptomes from single-cell RNA sequencing (scRNA-seq) of pancreatic cells, the only available data is the feature matrix. Using our method, we directly learn a coarsened graph from these features and perform downstream tasks on the coarsened graph. This approach demonstrates practical applicability and efficiency, particularly for datasets where graph structures are not explicitly provided.
>
> Next, the key contribution of this work lies in demonstrating how to perform graph-based dimensionality reduction and downstream tasks by leveraging informative features without relying on the graph structure. This approach offers several advantages:
>
> **1.Scalability Without Graph Structure:** It is naturally applicable in scenarios where the graph structure is unavailable, as it directly learns a coarsened graph.
>
> **2.Efficiency for Homophilic Graphs:** By omitting the graph structure, it simplifies implementation, making it more efficient and faster for homophilic datasets.
>
> **3.Robustness to Noisy Graphs:** It serves as a natural choice when the graph structure is noisy, ensuring reliable outcomes.
>
> [1] Kalofolias, Vassilis. "How to learn a graph from smooth signals." Artificial intelligence and statistics. PMLR, 2016.
>
> [2] Kumar, S., Ying, J., Cardoso, J. V. D. M., & Palomar, D. P. (2020). A unified framework for structured graph learning via spectral constraints. Journal of Machine Learning Research, 21(22), 1-60.
>
> **Ans Q2 and W2**We thank the reviewer for the question. This work is the first to directly learn a coarsened graph from the raw data or feature matrix alone. In our study, we have focused on the three most recent baselines that outperform existing graph coarsening methods. Due to space limitations in the table, we have selected these baselines as the most appropriate for comparison. However, if the reviewer would like us to include comparisons with any other specific baselines, we would be happy to consider them.
>
>
> **Ans Q3 and W3** We thank the reviewer for the insightful question. Existing state-of-the-art methods typically suffer from high time complexity because they rely on both the graph structure and the feature matrix. In contrast, our approach utilizes only the feature matrix, significantly reducing the time complexity of the algorithm.

---

> > ### Author Response · Authors · 2024-11-30
> > **reply to reviewer  ntd9**
> >
> > **Ans W3 and Q3** Our algorithm is capable of handling very large datasets, such as the OBGN product, which contains approximately 2.4 million nodes. In fact, many existing state-of-the-art algorithms encounter memory errors when applied to this dataset. While the FgC optimization-based method can run on this dataset, it requires a prohibitively long amount of time, as shown in Table 4 of the paper. This highlights the efficiency of our algorithm, which is able to process large datasets with far less computational overhead and less time complexity.
> >
> > **Ans W4 and Q4**  We thank the reviewer for the insightful question. As the optimization problem in our work is non-convex, we have utilized the **BSUM (Block Successive Upper Bound Minimization)** technique to develop our algorithm. This approach involves solving for one variable at a time while keeping the other variables fixed, which helps in addressing the non-convexity of the problem.
> >
> > 1. **Variable Update for  $w$**:
> >    To solve for $w$, we first **majorize** the objective function, creating an upper bound that simplifies the optimization process. Then, we solve the majorized problem, which allows us to derive a **closed-form solution** for \( w \). This step ensures that we can efficiently update $w$ without requiring iterative numerical methods.
> >
> > 2. **Update for $ P$**:
> >    For the update of $ P$, we utilize the **Karush-Kuhn-Tucker (KKT) conditions**, which provide a set of necessary conditions for optimality in constrained optimization problems. By applying the KKT conditions, we derive a straightforward update rule for $P$, making the optimization process more efficient and easier to compute.
> >
> > 3. **Update for  $\tilde{X}$**:
> >    The optimization problem for $\tilde{X}$ is relatively simple. By directly setting the gradient of the objective function with respect to $\tilde{X}$ to zero, we can easily obtain the solution for $\tilde{X}$
> >
> > We have applied the **BSUM technique** to solve the non-convex optimization problem, incorporating common and well-established techniques like majorization and the KKT conditions for efficient variable updates. This ensures that our approach is both computationally feasible and theoretically sound, providing a clear and efficient solution to the problem.
> >
> > If the reviewer believes that we have addressed all the reviews, we request you to please raise the mark.

---

> > > ### Author Response · Authors · 2024-12-02
> > > **Reply to rewier ntd9**
> > >
> > > Dear Reviewer,
> > >
> > > Thanks again for taking the time to review our paper and for your encouraging feedback! May we enquire if all the concerns you raised have been adequately addressed? Thank you very much.
> > >
> > > Best Regards, The authors

---

> ### Author Response · Authors · 2024-12-04
> **Reply to rewier ntd9**
>
> Dear Reviewer,
>
> The deadline for the author-reviewer discussion is approaching, and we wanted to kindly check if we have adequately addressed all your concerns. Please let us know if there are any remaining issues or clarifications needed from our side.

---

### Official Review · Reviewer_fvCX · 2024-11-04

**Soundness:** 4
**Presentation:** 4
**Contribution:** 3
**Rating:** 6
**Confidence:** 4

**Summary:**

This paper studies the scalability and structural limitations of existing graph coarsening techniques. It proposes a new framework, Coarsened Graph Learning (CGL), to directly learn a reduced graph from attribute data alone, eliminating the need for a pre-existing graph. By learning the graph from features, CGL enables scalable GNN training, is resilient against adversarial attacks, and incorporates semi-supervised learning with label information for enhanced downstream task performance. Experimental comparisons show that CGL outperforms state-of-the-art methods in node classification accuracy and computational efficiency across various datasets, proving its potential in large-scale, real-world applications.

**Strengths:**

1. The method relies solely on node features and labels, achieving impressive performance even without an initial graph structure. This approach shows potential for unifying graph data with other data formats.

2. This method stands out for its efficiency and resilience against structural attacks.

3. Bridging the sparsity of PY with the homophily of coarsening introduces an innovative and promising concept.

**Weaknesses:**

1. Since validation and test labels should remain hidden during training, it would be helpful to clarify how they are masked, perhaps by introducing a specific notation or symbol for this purpose.

2. Some baseline results are not fully reproduced. For instance, GCond typically produces results close to those of the full dataset, suggesting that the authors may not have adjusted the dataset split to 80%/10%/10% when replicating GCond’s performance.

3. Testing this method on large heterophilous graphs, such as Penn94, would add valuable insights into its scalability and effectiveness in diverse graph structures.

4. This method shares similarities with FGC [1] in objective design and learning approach. However, a more in-depth methodological comparison between the two would be beneficial for understanding their differences and relative strengths. Adding a dedicated section on related works to systematically compare various graph coarsening and condensation methods [2] would further enhance the paper.


### References
[1] Featured graph coarsening with similarity guarantees. ICML 2023

[2] A Comprehensive Survey on Graph Reduction: Sparsification, Coarsening, and Condensation. IJCAI 2024

**Questions:**

Although the runtime for each experiment is very fast, this method depends heavily on extensive hyperparameter tuning. Do the authors have any suggestions for how to select the hyper-parameters?

---

> ### Author Response · Authors · 2024-11-30
> **Reply to Reviewer fvCX**
>
> We sincerely thank the reviewers for their valuable feedback that we have used to improve the quality of our manuscript. Point-by-point responses are listed below.
>
> **Ans W1 and W2** Thank you for the insightful question. We implemented the masking process as follows: First, we apply one-hot encoding to the labels of the original graph. Next, we randomly set 20% of the rows to zero, allocating 10% of the nodes for testing and 10% for validation, and represent this matrix as \(Y\).
>
> After masking the labels, some of the label information (i.e., matrix \(Y\)) is used to derive the labels for the coarsened graph. This is achieved using the relation $\tilde{Y} = \arg\max(PY)$, where $\arg\max$ assigns a label to each supernode based on the label most frequent among the nodes within that supernode. The coarsened graph, now labelled, is then used to train the neural network. Testing is performed on the original graph, ensuring that the model generalizes well to the unmodified data.
>
> Additionally, we applied the same masking procedure for all experiments, including those for existing techniques such as GCOND, FGC, and SCAL. These methods utilize different levels of masking, but to ensure a fair comparison, we employed the same masking settings for both the existing methods and our proposed approach.
>
> **Ans W3** We thank the reviewer for the suggestion. We have performed experiments on the heterophilic dataset Penn 94. Below is the table showing the results of the experiments on the Penn 94 dataset as suggested by the reviewer:
>
> | Dataset   | Coarsening Ratio | GCOND  | SCAL  | FGC   | SCGL  |
> |-----------|------------------|--------|-------|-------|-------|
> | Penn 94   | 0.05             | 59.23  | 55.45 | 57.45 | 64.32 |
> | Penn 94   | 0.03             | 58.35  | 52.78 | 57.58 | 63.28 |
> | Penn 94   | 0.01             | 58.28  | 51.56 | 56.47 | 62.65 |
>
> This table demonstrate that the proposed SCGL technique outperforms the existing state of the art techniques.
>
> **Ans W4**  We thank the reviewer for suggestion. We have added a subsection titled **Graph Dimensionality Reduction** in Section 2 of the revised manuscript. Additionally, we would like to clarify that both FGC and the proposed SCGL framework are optimization-based techniques for graph coarsening. However, the formulations, objectives, inputs and output of the two algorithms are different.
>
> In FGC, given an original graph $\mathcal{G}(X, L)$, where $X$ is the feature matrix and $L$ is the Laplacian matrix of the original graph, the algorithm aims to learn a mapping matrix $C$ and each non-zero entry in the $C$ matrix indicates that the $i$-th node of the original graph is mapped to the $j$-th supernode of the coarsened graph. Once the matrix $C$ is obtained, the Laplacian of the coarsened graph is computed using the relation $L_c = C^T L C$. The time complexity of the FGC algorithm is $\mathcal{O}(p^2 k)$, where $p$ is the number of nodes in the original graph and $k$ is the number of nodes in the coarsened graph.
>
> In contrast, the proposed SCGL technique aims to learn a coarsened graph directly from the raw feature matrix $X$, without explicitly considering the graph structure of the original graph. The time complexity of the proposed CGL algorithm is $\mathcal{O}(k^2 p)$, where $k$ is the number of nodes in the coarsened graph and $p$ is the number of nodes in the original graph. Moreover, the key contribution of this work lies in demonstrating how to perform graph-based dimensionality reduction and downstream tasks by leveraging informative features without relying on the graph structure. This approach offers several advantages:
>
> **1.Scalability Without Graph Structure:** It is naturally applicable in scenarios where the graph structure is unavailable, as it directly learns a coarsened graph.
> **Efficiency for Homophilic Graphs:** By omitting the graph structure, it simplifies implementation, making it more efficient and faster for homophilic datasets.
> **Robustness to Noisy Graphs:** It serves as a natural choice when the graph structure is noisy, ensuring reliable outcomes.
>
> **Ans Q1** Thank you for the reviewer’s suggestion. To accelerate the hyperparameter tuning process, we have employed a Bayesian optimization strategy. This approach efficiently explores the hyperparameter space, leveraging prior knowledge to identify optimal configurations with fewer evaluations, thereby significantly reducing the computational cost of tuning.
>
>
> If the reviewer believes that we have addressed all the reviews, we request you to please raise the mark.

---

> > ### Author Response · Authors · 2024-12-02
> > **Reply to Reviewer fvCX**
> >
> > Dear Reviewer,
> >
> > Thanks again for taking the time to review our paper and for your encouraging feedback! May we enquire if all the concerns you raised have been adequately addressed? Thank you very much.
> >
> > Best Regards, The authors

---

> ### Author Response · Authors · 2024-12-04
> **Reply to Reviewer fvCX**
>
> Dear Reviewer,
>
> The deadline for the author-reviewer discussion is approaching, and we wanted to kindly check if we have adequately addressed all your concerns. Please let us know if there are any remaining issues or clarifications needed from our side.

---

### Meta-Review · Area_Chair_P2ZC · 2024-12-21

**Metareview:**

The paper proposes a new framework, Coarsened Graph Learning (CGL), to directly learn a reduced graph from attribute data and address the scalability and structural limitations of existing graph coarsening techniques. The problem of graph coarsening is important. The proposed model has lower computational complexity compared to baselines. The presentation is clear. Several key issues need to be addressed, including insufficient related work discussion and comparison, unclear motivation of method design, and insufficient experiments. Reviewers are generally negative about this work.

**Additional Comments On Reviewer Discussion:**

I posted a discussion, but no one replied. Reviewers are generally negative about this work.

---

### Decision · Program_Chairs · 2025-01-22

Reject